# Accessibility to rabies centers and human rabies post-exposure prophylaxis rates in Cambodia: A Bayesian spatio-temporal analysis to identify optimal locations for future centers

Jerome N. Baron[1,¤a]*, Véronique Chevalier[2,3,4,¤b], Sowath Ly[4], Veasna Duong[5], Philippe Dussart[5,¤c], Didier Fontenille[6,¤d], Yik Sing Peng[4], Beatriz Martínez-López[1]

1 Center for Animal Disease Modeling and Surveillance (CADMS), Department of Medicine & Epidemiology, School of Veterinary Medicine, University of California, Davis, California, United States of America, 2 CIRAD, UMR ASTRE, Phnom Penh, Cambodia, 3 ASTRE, Univ Montpellier, CIRAD, INRA, Montpellier, France, 4 Epidemiology and Public Health Unit, Institut Pasteur du Cambodge, Phnom Penh, Cambodia, 5 Virology Unit, Institut Pasteur du Cambodge, Phnom Penh, Cambodia, 6 Institut Pasteur du Cambodge, Phnom Penh, Cambodia

¤a Current address: Section for Epidemiological Methods, Department of Epidemiology and Disease Control, National Veterinary Institute of Sweden, Uppsala, Sweden
¤b Current address: Epidemiology Unit, Institut Pasteur de Madagascar, Antananarivo, Madagascar
¤c Current address: Institut Pasteur de Madagascar, Antananarivo, Madagascar
¤d Current address: MIVEGEC, Université de Montpelier, CNRS, IRD, Montpelier, France
* jbaron@hotmail.fr

**Data Availability Statement:** Original data is proprietary to the Pasteur Institute of Cambodia.

## Abstract

Rabies is endemic in Cambodia. For exposed humans, post-exposure prophylaxis (PEP) is very effective in preventing this otherwise fatal disease. The Institut Pasteur du Cambodge (IPC) in Phnom Penh was the primary distributor of PEP in Cambodia until 2018. Since then, and to increase distribution of PEP, two new centers have been opened by IPC in the provinces of Battambang and Kampong Cham. Data on bitten patients, who sometimes bring the head of the biting animal for rabies analyses, have been recorded by IPC since 2000. However, human cases are not routinely recorded in Cambodia, making it difficult to establish a human burden of disease and generate a risk map of dog bites to inform the selection of future PEP center locations in high-risk areas. Our aim was to assess the impact of accessibility to rabies centers on the yearly rate of PEP patients in the population and generate a risk map to identify the locations where new centers would be the most beneficial to the Cambodian population. To accomplish this, we used spatio-temporal Bayesian regression models with the number of PEP patients as the outcome. The primary exposure variable considered was travel time to the nearest IPC center. Secondary exposure variables consisted of travel time to a provincial capital and urban proportion of the population. Between 2000 and 2016, a total of 293,955 PEP patient records were identified. Our results showed a significant negative association between travel time to IPC and the rate of PEP patients: an increase in one hour travel time from the living location to IPC PEP centers leads to a reduction in PEP rate of 70% to 80%. Five provinces were identified as the most

This data contains personal individual patient information and cannot be shared as such. An aggregated and de-identified version of the dataset is shared within the manuscript and supporting information files.

**Funding:** This survey has been jointly granted by the Swedish Research Council and Region Occitanie. (VC) Funding for the PhD student JNB conducting the analysis was provided for the academic years 2019-20 and 2020-21 by the UC Davis School of Veterinary Medicine Graduate Student Support Program (GSSP) fellowship. GSSP funding for 2020-21 was provided through the McKay endowment. (https://www.vetmed. ucdavis.edu/research/student-research/gssp/gssp-funding). The funders had no role in study design, data collection and analysis, decision to publish, or preparation of the manuscript.

**Competing interests:** The authors have declared that no competing interests exist.

efficient locations for future centers to maximize PEP accessibility: Banteay Meanchey, Siem Reap, Takeo, Kampot and Svay Rieng. Adding a PEP center in every provincial capital would increase the proportion of Cambodians living within 60 minutes of a PEP center from 26.6% to 64.9%, and living within 120 minutes from 52.8% to 93.3%, which could save hundreds of lives annually.

## Author summary

Rabies is a fatal viral disease that affects the nervous system. It is endemic in many countries in Africa and Asia where free roaming dogs form a reservoir. Transmission to humans occurs most often through a dog bite. However, post-exposure prophylaxis (PEP), if administered before symptom onset, is highly effective at preventing the disease. In Cambodia, a few number of centers offer PEP, with the Institut Pasteur du Cambodge in Phnom Penh being the main one. These few locations lead to limited accessibility for rural areas distant from Phnom Penh and underestimations of the dog-bite burden and PEP needs. Through statistical modelling, we measured the impact of accessibility on the number of PEP patients and predicted the impact of opening new centers in other locations. We found that travel time was significantly associated with the rate of PEP patients. IPC opened new rabies centers in Battambang and Kampong Cham provinces in 2018 and 2019, respectively, and we identified four provinces where future openings would be the most beneficial: Banteay Meanchey, Siem Reap, Takeo, Kampot and Svay Rieng. This study is part of a broader drive to eradicate rabies in Cambodia by 2030 through increased PEP infrastructure and control measures in the dog population.

## 1. Introduction

Rabies is a neurotropic virus and a member of the genus Lyssavirus in the Rhabdoviridae family, for which multiple carnivore species are reservoirs [1,2]. Most mammals can be infected, including humans and common domestic species [3]. It is of great concern to humans both in terms of health and economic impacts with nearly 59,000 people dying from rabies every year and an overall cost of $8.6 billion, including direct and indirect costs from human cases and loss of livestock [4,5]. The societal cost is compounded by the violent nature of the disease, the absence of treatment for symptomatic patients, and the young age of victims, with a median age of 26 in Cambodia [6]. The virus is transmitted through the saliva of an infectious animal, most commonly through a bite. The incubation period is typically three to eight weeks, but can sometimes be months, and the disease is nearly 100% fatal once clinical symptoms appear [7,8]. The vast majority of cases occur in developing countries in Africa and Asia with nearly 99% of human cases resulting from dog bites. Eighty seven percent of these cases occur in rural areas [3]. Cambodia is one such country where rabies is endemic and has one of the highest incidence rates of human rabies in the world, with an estimated 6 cases per 100,000 people annually, representing approximately 800 cases and more than 375,000 dog bite injuries per year [6,9,10].

Rabies vaccines have proven very effective for protecting humans, dogs, and wildlife, but the use of the vaccine is highly variable globally. Moreover, even in the event of human exposure, most cases can be avoided with timely administration of post-exposure prophylaxis (PEP) [11]. Since 2018, the WHO has recommended a one-week PEP regimen involving the

administration of human rabies immune globulins (HRIG) in combination with a 2-site intradermal (ID) injection of the first dose of rabies vaccine, ideally on the day of exposure. Two subsequent vaccine doses are administered 3 and 7 days after the first [12,13]. However, other longer regimens in use can be intramuscular (IM) or ID, 1 or 2-site, and add a fourth and sometimes fifth dose between 14 to 28 days after the first [11,12,14]. In comparison, typical rabies pre-exposure prophylaxis (PrEP) involves three vaccine doses on days 0, 7 and 21 [15]. With these tools and effective control and management of dog populations, rabies has been controlled and nearly eradicated in most of Europe and the Americas [16]. In Southeast Asia, without achieving full elimination, the situation improved with canine vaccination and expanded access to PEP, especially in Vietnam and Thailand, seeing a dramatic reduction in human and dog cases [17–20]. However, in areas where rabies remains endemic, successful mitigation of human rabies requires rapid and reliable availability of PEP for bite victims, in addition to effective and sustained dog vaccination strategies and appropriate dog population management.

Institut Pasteur du Cambodge (IPC), located in Cambodia's capital Phnom Penh, has made PEP available since 1998, with an average of 21,000 regimens being administered each year [9]. IPC has used the WHO-recommended one-week PEP regimen since 2018 [21]. PEP administration is recorded alongside patient interviews about the characteristics of the exposure. In addition, an ongoing passive surveillance program tests head samples from suspected animals responsible for bite injuries using PCR [22]. This program relies on patients voluntarily bringing biting animal heads when coming to IPC to seek PEP. However, with limited travel infrastructure, travelling to the capital can take precious time resulting in prohibitive loss of income and travel costs. Additionally, the cost of PEP varies from $30 to $70 in a country where the median monthly income is $100 [23]. Field surveys have also shown a relatively low awareness of rabies in rural areas, its mode of transmission, and the ways to prevent it following exposure, leading to many people not seeking proper care or preferring traditional medicine [24,25]. Thus, the centralized nature of IPC in Phnom Penh associated with the cost and lack of information means that access to this critical care is unevenly distributed across the country and that the rabies burden and the need for PEP is likely underestimated, especially in rural regions away from the capital. A few other locations such as the Angkor Children Hospital in Siem Reap and the National Institute of Public Health clinic in Phnom Penh also offer PEP, as well as a few private clinics. However, the quality and cost of PEP regimens at these private clinics remains unknown and a large majority of recorded PEP patients come to IPC [26]. In addition to the Phnom Penh PEP center, IPC opened two new locations for PEP treatment, one in Battambang city in July 2018, located in the far northwest of the country, and the second one in Kampong Cham, the third largest city of Cambodia, located 120km from Phnom Penh on the right bank of the Mekong River, in March 2019. Unfortunately, there is no national systematic surveillance for human cases and the passive surveillance for dog rabies is biased by the centralized nature of IPC, with most tested animals being from areas close to Phnom Penh. Thus, the direct assessment of human or animal rabies cannot be established to guide risk-based allocation of future PEP clinics. However, we can guide the geographic allocation of resources through indirect means from human PEP data. Using methods that better control for spatial heterogeneity and missing data whilst adjusting for the geographical accessibility (subsequently referred to as accessibility) to PEP centers, we can provide a more detailed picture of the rabies burden and the PEP needs in Cambodia.

A number of studies have successfully used Bayesian statistics to create disease risk maps in settings where case data is either unequally distributed, incomplete, or both, as is often the case with veterinary and neglected zoonotic diseases [27–30]. These methods can also fit complex regression model structures that can include spatial and temporal autocorrelation [31–41].

The primary goal of this paper was to assess the impact of accessibility to a PEP center and urbanization level of the province or district of origin on the observed number of PEP patients using Bayesian regression modeling. This will help to identify provinces where new PEP centers would be the most beneficial to the Cambodian population by investigating the potential impact of opening new PEP centers on expected numbers of PEP patients. This study is part of a broader effort to control rabies in Cambodia, helping to determine resource allocation, risk-based strategies, and guide policies to meet eradication targets.

## 2. Methods

We considered three variables potentially influencing the access to the PEP centers for bite victims: travel time to the nearest PEP center, travel time to the closest provincial capital, and urban population proportion as a proxy for socio-economic status and accessibility to general healthcare and health information.

### 2.1 Data collection and management

**2.1.1 Rabies surveillance data.** Since 1998, IPC has made PEP available to dog-bite victims at its Phnom Penh location, initially for free, and then with a fee starting in 2010 [23]. In parallel, each patient was given a questionnaire upon arrival to collect information related to the characteristics of the attack and the victim(s), such as name, age, and address, in order to guide allocation of HRIG and the PEP regimen. These data were then completed over time with follow-up visit information, number of injections, and results from animal testing when the head of the biting animal was brought with patients. The victim's residence was recorded at the province level from 1998 to 2013, then at the district level from 2013 to 2016 (S1 and S2 Data).

**2.1.2 Administrative and demographic data.** Cambodia has four main levels of administrative divisions. The largest division is the province, followed by the district, the commune, and the village. Demographic data (population size and urban population proportion) by province and district where obtained from the 1998, 2008 and 2019 official population censuses of Cambodia [42–44]. A linear population growth was assumed and projected between the 1998 and 2008 census population values to estimate population values from 2000 to 2007. The same method was used between the 2008 to 2019 census population values to estimate values from 2009 to 2016 (S1 Fig and S3 Data).

Administrative maps were obtained from open-sourced data platforms [45,46]. We performed our modelling analyses at both province and district levels. Certain areas, such as Phnom Penh, have special nomenclature but follow the same overall structure in their divisions. Cambodia has undergone numerous administrative changes within the time frame of our study. The most important were: 1) the split of Kampong Cham Province into 2 provinces in 2013, creating the new province of Tboung Khmum; 2) the absorption of a number of densely populated communes from Kandal province into Phnom Penh Province causing the administrative transfer of approximatively 250,000 people between the two provinces; 3) the creation of numerous new districts across the country, increasing their number from 183 in 1998 to 204 by 2019. We standardized the population data to arbitrary fixed time points in terms of administrative divisions. For the province-level modelling, the 2010 map was used, which had the same administrative divisions as the 2008 census, i.e. 24 provinces and 192 districts. For the district-level modelling, the 2016 map comprising 25 provinces and 197 districts was used. Population totals for all years were adjusted as described below to take into account the administrative geography of the selected maps. For the province-level model, population values of the 2019 census for Tboung Khmum and Kampong Cham Provinces were aggregated

into the single Kampong Cham Province to reflect the 2010 map boundaries. The same was done for IPC data for the years 2013 to 2016. We identified individual communes that were transferred from Kandal Province to Phnom Penh to adjust projections when the transfer was made in 2013. Thus, projections from 2008 to 2012 assumed the administrative division of 2008, and projections from 2013 to 2017 assumed the administrative divisions of 2016. Individual patient records at the district level were manually cross-checked using province and commune names to ensure the information matched the administrative map in use.

The urban proportions of provinces and districts were calculated from the definition of urban communes used in the 1998 and 2008 censuses. This definition was modified in the 2019 census; however we kept using the older standard for consistency in the projections.

**2.1.3 Accessibility data.** We used the global friction raster from the Malaria Atlas Project to create a variable measuring accessibility to PEP centers and provincial capitals from the patient's residence [47]. Using primary data covering transport infrastructure, land coverage, geographical features such as slope, altitude, and water bodies, this dataset estimates the travel speeds across the globe with a resolution of one square kilometer. Using the coordinates of PEP centers or provincial and district capitals and the friction raster, the R package "gdistance" computes the shortest travel time from any pixel on the map to the nearest point (PEP center or provincial/district capital) by assessing all possible paths between them [48,49]. This produces a raster of travel time from any given location to the nearest point with the same resolution as the friction raster.

**2.1.4 Prediction scenarios for PEP patients.** We considered five accessibility scenarios for predicting the expected number of PEP patients (Table 1). The first scenario (Scenario 1) represents the situation that was in place from the post-war re-opening of IPC in 1995 with one PEP center in Phnom Penh until 2017 (S2A Fig). This scenario was used to construct the statistical model as it represents the situation in place when the IPC surveillance data were collected from 2000 to 2016. This scenario was also used to predict patient numbers if no new centers were opened after 2016. The second scenario (Scenario 2) incorporates 3 PEP centers: the IPC Phnom Penh center as well as the two new PEP centers, one at the Department of Health facility in the city of Battambang, which opened in 2018, and

**Table 1. Description of accessibility scenarios used for model fitting and predictions of future PEP patient rates and numbers.**

| | Description | Source | Model usage | Spatial scale of model |
|---|---|---|---|---|
| Scenario 1 | One PEP center available in Phnom Penh. This represents the situation in place from 1995 to 2017 and includes the data collection period from 2000 to 2016. | Coordinates of IPC were used to create accessibility raster | Model fitting and 2017 predictions | Province and district |
| Scenario 2 | Three PEP centers available in Phnom Penh, Battambang and Kampong Cham. This represents the situation in place currently, following the opening of two new IPC centers in Battambang city (2018) and Kampong Cham city (2019). | Coordinates of all three centers were used to create accessibility raster | 2017 predictions | Province and district |
| Scenario 3 | Theoretical scenario: a PEP center present in each provincial capital. | A spatial point data set of the centroid of provincial capitals was included with the 2010 administrative boundaries shapefiles [49] | 2017 predictions | Province and district |
| Scenario 4 | Theoretical scenario: a PEP center present in each district capital. | A spatial point data set of the centroid of district capitals was included with the 2010 administrative boundaries shapefiles [49]. Provincial capitals are also the administrative centers of their own district. | 2017 predictions | Province and district |
| Scenario 5 | 21 scenarios of four PEP centers which include the three current PEP centers (Phnom Penh, Battambang and Kampong Cham) and one additional center added in each of the 21 provincial capitals. | as scenario 1, 2 and 3 | 2017 predictions | District |

the other at the Kampong Cham Hospital, which opened in 2019 (S2B Fig). This scenario was used to predict the numbers of patients based on the PEP access available in Cambodia from 2019 through 2020. The three last scenarios represented a situation with theoretical openings of new PEP centers (S2C and S2D Fig). The first one assumed a PEP facility in every provincial capital (Scenario 3) and the other assumed a PEP facility in every district capital (Scenario 4). As Scenario 3 assumed the un-realistic simultaneous addition of centers in every provincial capital at the same time, a set of multiple simulations (Scenario 5) adding a center in a single provincial capital at a time (in addition to the three centers already in existence) were used to observe the specific impact of each location and identify locations where a new center would maximize the number of new patients. Scenario 5 was run with the district level model as earlier models showed that the higher spatial resolution better captured the population distribution in regards to accessibility and was considered more accurate for advising the location of future centers. The accessibility measure was aggregated by means of extracting the median travel time to a PEP center or provincial/district capital in each province or district. These provinces and districts could then be linked to the patient's province or district of residence.

## 2.2 Statistical analysis

**2.2.1 Accessibility descriptive analysis.** For each of the five accessibility scenarios, we used our 2016 demographic projections at the district level and the aggregated accessibility data to estimate the proportion of the population living in districts that were within 60 minutes and 120 minutes travelling time to the nearest PEP center.

**2.2.2 PEP models.** To estimate the impact of accessibility on PEP patient rates and predict new patients, we applied a Bayesian modelling framework using Integrated Nested Laplace Approximations (INLA). Analyses were performed with the R package "R-INLA" [50–53]. We investigated two spatial scales (province and district) and two temporal scales (year and month). The predicted PEP patient numbers were modelled using a Poisson regression with a population offset in which time was used as a non-parametric autocorrelated temporal effect. Three fixed effects were considered: travel time to the closest PEP center, urban population proportion of the district or province of the patient's residence, and travel time to the closest provincial capital. The graphical representation did not suggest any seasonal pattern. This was confirmed by a preliminary analysis of the number of patients every month using a generalized additive model (GAM) with a smooth term over month, using the R-package "mgcv" [54] (S3 Fig). The smooth term for month from this model had no significant effect on the number of PEP patients (p-value = 0.815).Therefore, we constructed all prediction models using a year-level temporal random effect only. Model selection was done using the Deviance Information Criterion (DIC) and the Watanabe-Akaike Information Criterion (WAIC). All models in this section and beyond used minimal non-informative priors as set by default in R-INLA. The default prior distribution for a Poisson model in R-INLA is a Gamma distribution with the following parameters $Gam(1, 0.00005)$ [55]. Once the best fitting model was obtained, both at the provincial and district level, models were fitted using a dataset that was expanded to include the prediction year 2017. This involved adding observations with demographic and accessibility data for 2017 but with NA values for the outcome, to be fitted when running the model. Due to the different spatial aggregation scales, resulting models were not directly comparable with DIC and WAIC. Therefore, we used intra-class correlation coefficients (ICC) to assess the agreement between observed and fitted values. These ICCs where computed using the R package "irr" [56].

All data management and statistical analysis were conducted in R Version 4.0.3 [25].

### 3. Results

#### 3.1 Descriptive data

Between 2000 and 2016, 293,955 PEP patient records were identified and associated to a given province, representing a rate of 12.97 patients per 10,000 person-years. For another 85 records (0.03%) the province could not be identified. From 2013 to 2016 we observed 85,780 records with an identifiable district, whereas a further 42 patients (0.05%) could not be located. Yearly data are summarized in Fig 1. We observed two phases in PEP patient numbers with a plateau between 12,000 and 14,000 patients per year from 2000 to 2007, followed by a major increase to a second plateau between 20,000 and 22,000 patients per year from 2008 to 2016. Despite the Cambodian population increase during this time, the rate of PEP per 10,000 people also showed a major increase from 2007 to 2008.

The distribution of the geographic origins of PEP patients was very heterogeneous, with 158,009 patients (53.8%) coming from the capital city of Phnom Penh and another 60,267 (20.5%) coming from the province of Kandal, which surrounds Phnom Penh (Fig 2). In comparison, seven provinces had less than 100 PEP patients in the 17 years of data, and another seven provinces had between 100 and 1,000 patients. Phnom Penh and Kandal also had the highest rates of patient recruitment with 65.3 and 29.3 patients per 10,000 person-years, respectively. The patient recruitment of all other provinces was below 10 (S1 Table). These patterns were also visible at the district level, with a ring of districts with high patient numbers

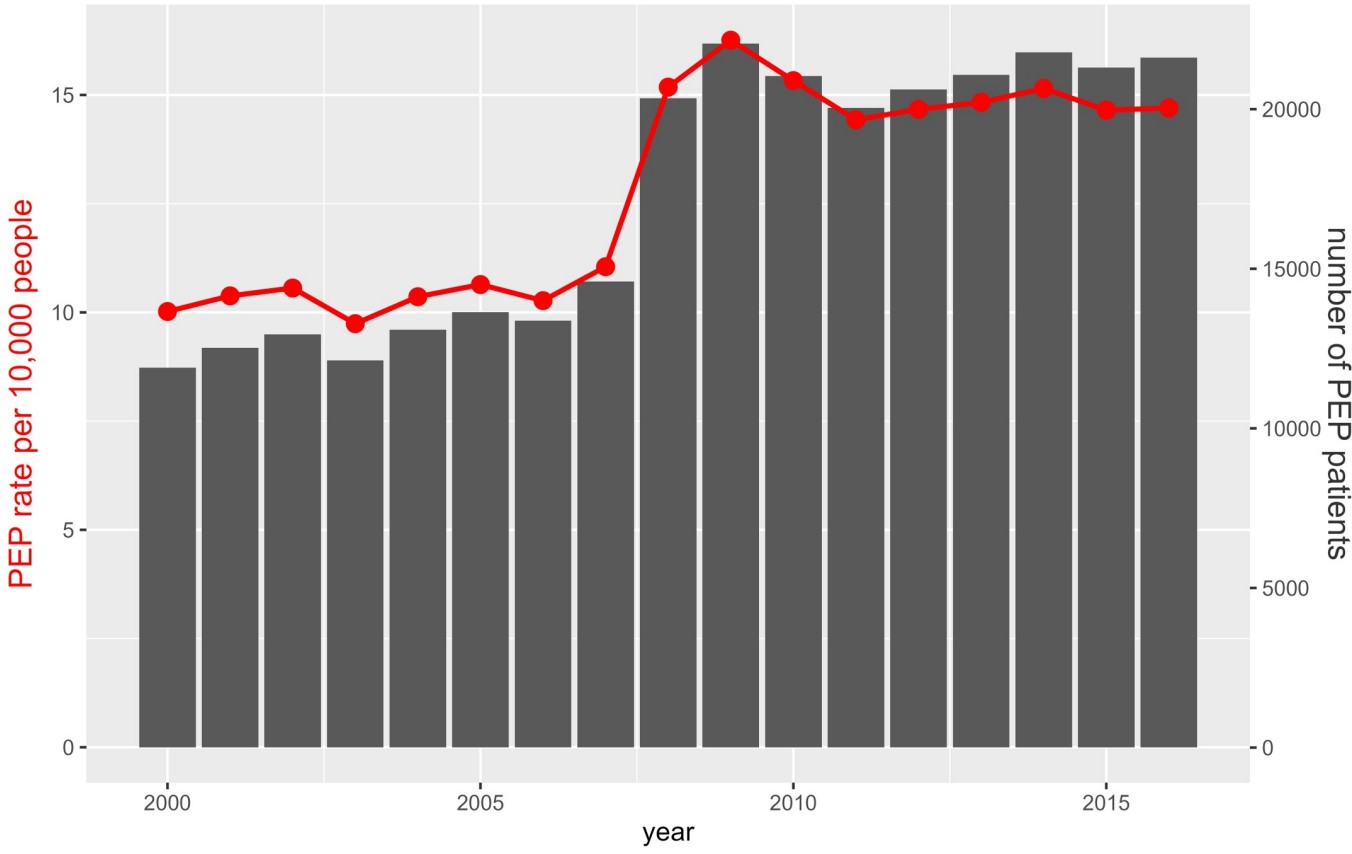

**Fig 1. Observed number and rates of PEP in Cambodia from 2000 to 2016.** Red curves represent rates per 10,000 people and histogram bars represent absolute numbers.

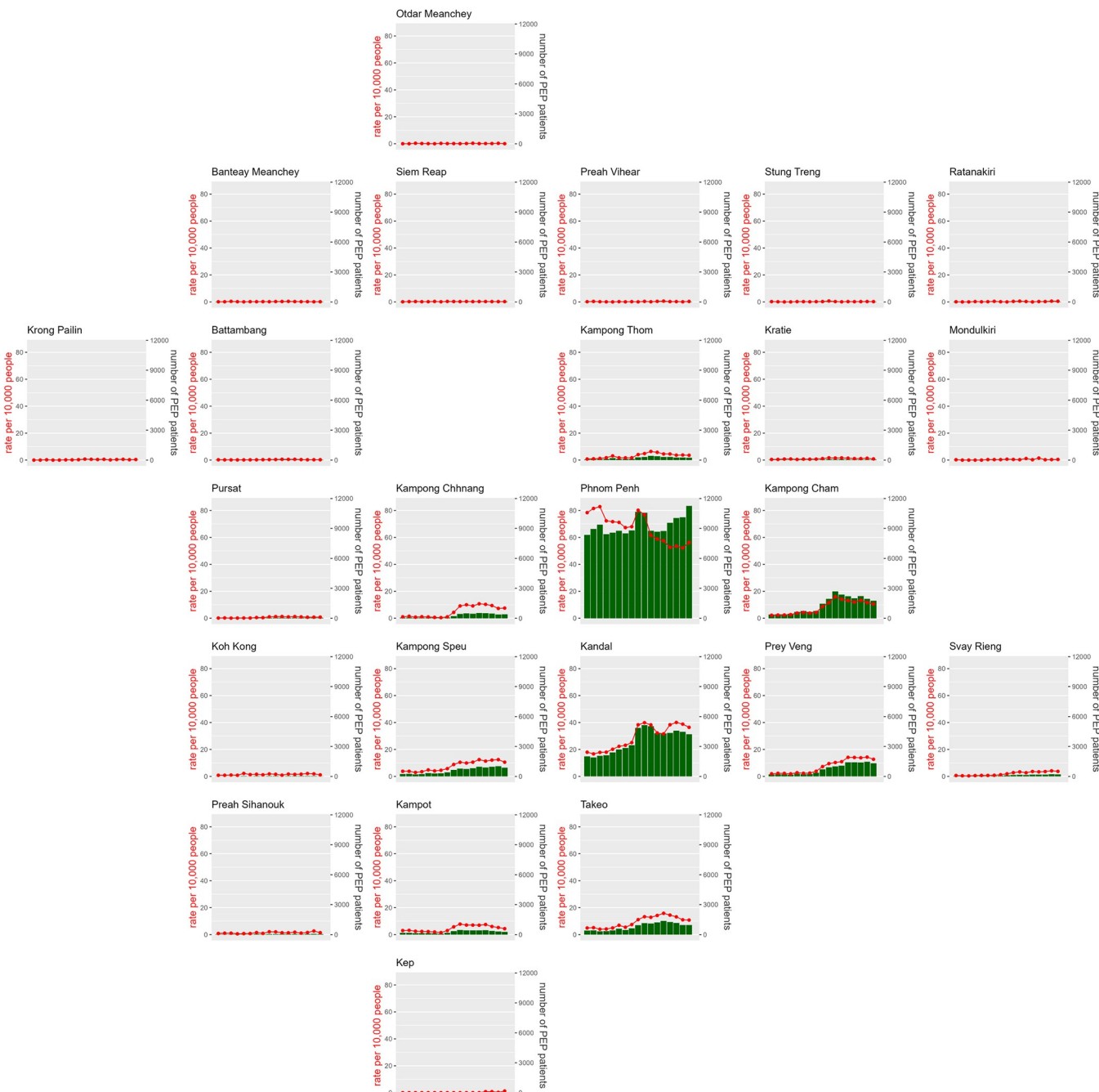

**Fig 2. Time series of PEP patients numbers and rates by province.** Red curves represent rates per 10,000 people and histogram bars represent absolute numbers.

and rates surrounding Phnom Penh, and very low values further away, with 13 districts without any patients between 2013 and 2016 (Fig 3).

## 3.2 Population accessibility

Based on our population projections, Cambodia had a population of 14.70 million people in 2016. Of these, 26.6% lived in districts with a median travel time of 60 minutes or less to the

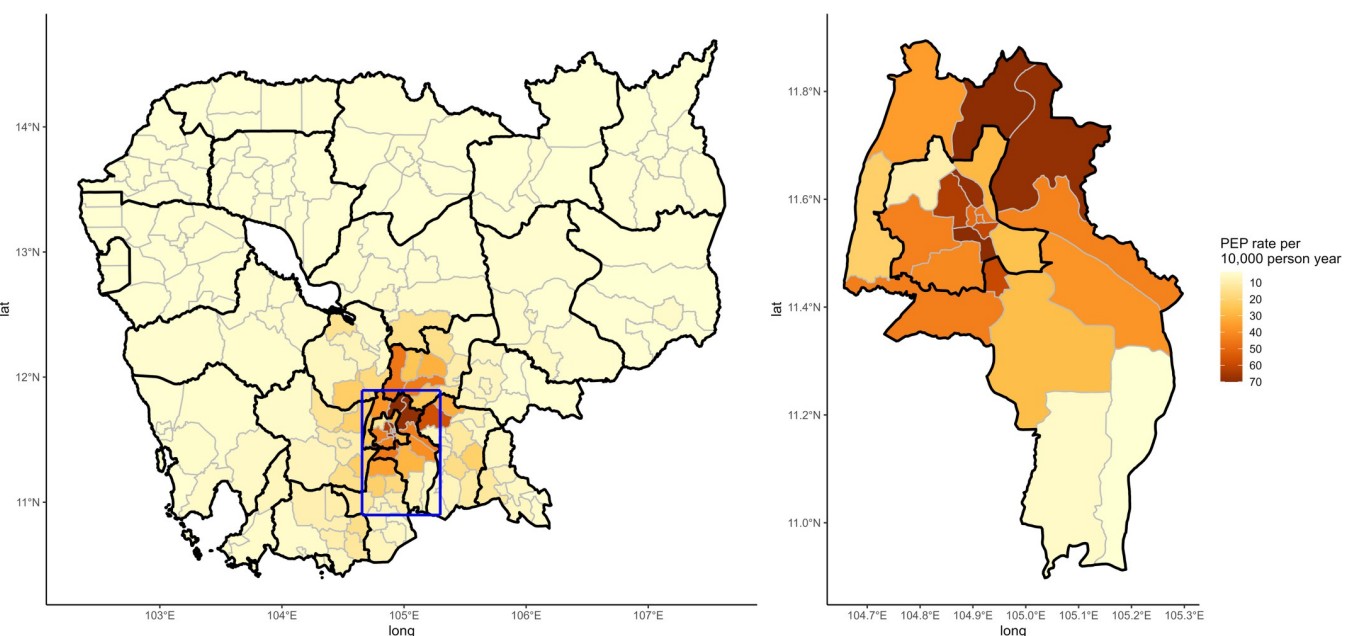

**Fig 3. Observed average rates of PEP patients per district for the years 2013 to 2016.** The inlet focuses on the provinces of Kandal and Phnom Penh where the majority of PEP patients at IPC come from. Bold lines represent provincial boundaries. Base map can be found at [45]. https://data.humdata.org/dataset/cambodia-admin-level-0-international-boundaries. Details for the corresponding license can be found at: https://data.humdata.org/faqs/licenses.

IPC PEP center in Phnom Penh, and another 26.2% lived in districts with a median travel time between 60 and 120 minutes of that center, for a total of 52.8% within 120 minutes. In comparison, from 2013 to 2016 74.9% of PEP patients came from districts within 60 minutes of IPC and another 21.5% between 60 and 120 minutes. Opening the two new centers in Battambang and Kampong Cham Provinces (Scenario 2) resulted in 36.6% of the population living within 60 minutes of a PEP center and 34.8% living between 60 and 120 minutes, bringing the total to 71.5% within 120 minutes from a PEP center. Testing the location of new provincial PEP centers individually in each of the remaining provinces (Scenario 5) increased the proportion of the population living within 60 minutes of a PEP center up to 41.3% with a center added in Svay Rieng Province and 78.3% within 120 minutes with a PEP center added in Siem Reap Province (S2 Table). The least impactful scenarios were adding a center in Kandal or Mondul Kiri Provinces which increased the population living within 60 minutes of a PEP center by 0% and 0.1%, respectively and within 120 minutes by 0.3% and 0.1%, respectively. The four provinces where adding a PEP center had the largest accessibility impact were Banteay Meanchey, Siem Reap, Svay Rieng, Takeo, each adding more than 500,000 people living within 60 minutes of a PEP center. Another three provinces, Kampot, Prey Vaeng and Pursat, added between 315,000 and 380,000 people living within 60 minutes of a PEP center. The remaining provinces yielded values below 215,000. Adding a PEP center in every provincial capital would increase the proportion of Cambodians living within 60 minutes of a PEP center to 64.9% and to 93.3% within 120 minutes.

### 3.3 PEP model results

At both the province and district levels, the PEP patient model was fitted with three variables: travelling time to the closest IPC PEP center, urban population proportion, and travel time to the provincial capital. At both province and district level, and in both univariate and

**Table 2. Rate ratios for fixed effects (with 95% credibility intervals) in univariate and multivariate Bayesian Poisson regression models.**

| Fixed effect variable | Province level | | District level | |
|---|---|---|---|---|
| | Univariate | Multivariate | Univariate | Multivariate |
| Time to vaccination center (1h) | 0.275 (0.273 to 0.276) | 0.265 (0.261 to 0.279) | 0.297 (0.294 to 0.300) | 0.196 (0.192 to 0.199) |
| Time to provincial capital (1h) | 0.061 (0.061 to 0.062) | 2.334 (2.244 to 2.426) | 0.157 (0.154 to 0.160) | 2.934 (2.824 to 3.049) |
| Urban proportion (10%) | 1.329 (1.328 to 1.331) | 1.092 (1.090 to 1.094) | 1.168 (1.166 to 1.169) | 1.017 (1.015 to 1.019) |

multivariate models, travel time to the closest IPC PEP center had a significantly negative association with the rate of PEP patients. These rate ratios varied from 0.20 to 0.30 (Table 2). Thus, increasing travel time by 1 hour reduced PEP rates by a range of 70% to 80%, depending on the model. Time to provincial capital also had a strongly significant negative association with the rate of PEP in univariate models at both the province and district levels (RR = 0.06 and 0.16, respectively). However, once adjusted for the other two variables, this relationship became strongly positive in both cases (RR = 2.33 and 2.93, respectively). Finally, urban population proportion had a positive association with PEP rate in both univariate models (RR = 1.33 and 1.17, respectively). In both the province- and district-level models, adjusting for accessibility to PEP centers and provincial capital reduced the effect size of urban proportion, bringing the rate ratios closer to the null (rate ratio of 1) with values of 1.09 at the province level and 1.02 at the district level.

Coefficients for the random effect of year mostly followed the pattern of the observed rate of testing, with a plateau from 2000–2007, before a high increase in 2008 followed a slow reduction after 2008. The district-level model, which was fitted using 2013–2016 data, only shows the downwards trend of PEP rates. When comparing the ICC of the two multivariate models, we observe a closer fit of the data at the provincial level with an ICC of 0.97 (95% CI 0.96–0.97), compared to the district-level model with an ICC of 0.90 (95% CI 0.88–0.91).

### 3.4 Predictions for 2017

Predictions for 2017 had broadly similar topline numbers at the district and province levels. In Scenario 1, which corresponds to the initial situation with only one PEP center in Phnom Penh, the province-level model predicted 21,885 PEP patients for 2017, compared to 21,611 observed patients in 2016, and the district-level model predicted 21,643 PEP patients (S3 Table). To be able to compare both province and district level models, results from the district model were aggregated at the province level in the following descriptions.

The opening of the two new centers in Battambang and Kampong Cham Provinces (Scenario 2) led to predictions of 28,040 patients for the province-level model and 29,950 patients for the district-level model, which corresponded to an increase in the number of predicted patients of 6,155 and 8,307 respectively. In both models, the two provinces with the largest increases in numbers of patients were Battambang and Kampong Cham, where the new PEP centers were located. These combined provinces went from 1,067 to 5,290 predicted patients in the province-level model, which represented 69% of the national predicted increase. In the district-level model, the two provinces went from 1,321 to 5,457 predicted patients, representing 50% of the national predicted increase. The new province of Tbong Khmum, which is included as part of Kampong Cham in the province-level model, also experienced a major increase from 305 to 2,161 patients in the district-level model (22% of the national increase). Most other changes came from provinces bordering Battambang and Kampong Cham, with Banteay Meanchey experiencing the largest increase in the number of predicted patients in both models, from 19 to 972 at the province level, and from 7 to 1,055 at the district level. The

number of predicted patients slightly increased in non-neighboring provinces. The localized impact of these two new centers was very visible when mapped at the district level, with concentric rings of increasing PEP rates in districts closer to the PEP center (Fig 4A and 4B). Full provincial predictions are detailed in S4 Table.

The theoretical Scenarios 3 and 4 led to larger numbers of predicted patients. The province-level model predicted 42,992 patients with a PEP center in each province (Scenario 3) and 76,638 with a PEP center in each district (Scenario 4). The district-level model predicted 50,944 and 92,601 for these two scenarios, respectively. These new patients are much more widely distributed across the country, but rates seemed to remain higher in districts and provinces surrounding Phnom Penh as well as in provinces in the North West (Fig 4C and 4D).

For each scenario except Scenario 1, the district-level model had higher predicted patient numbers than the province-level model. Simulating the opening of a new center in each province individually (Scenario 5) using the district level model led to increases in the number of

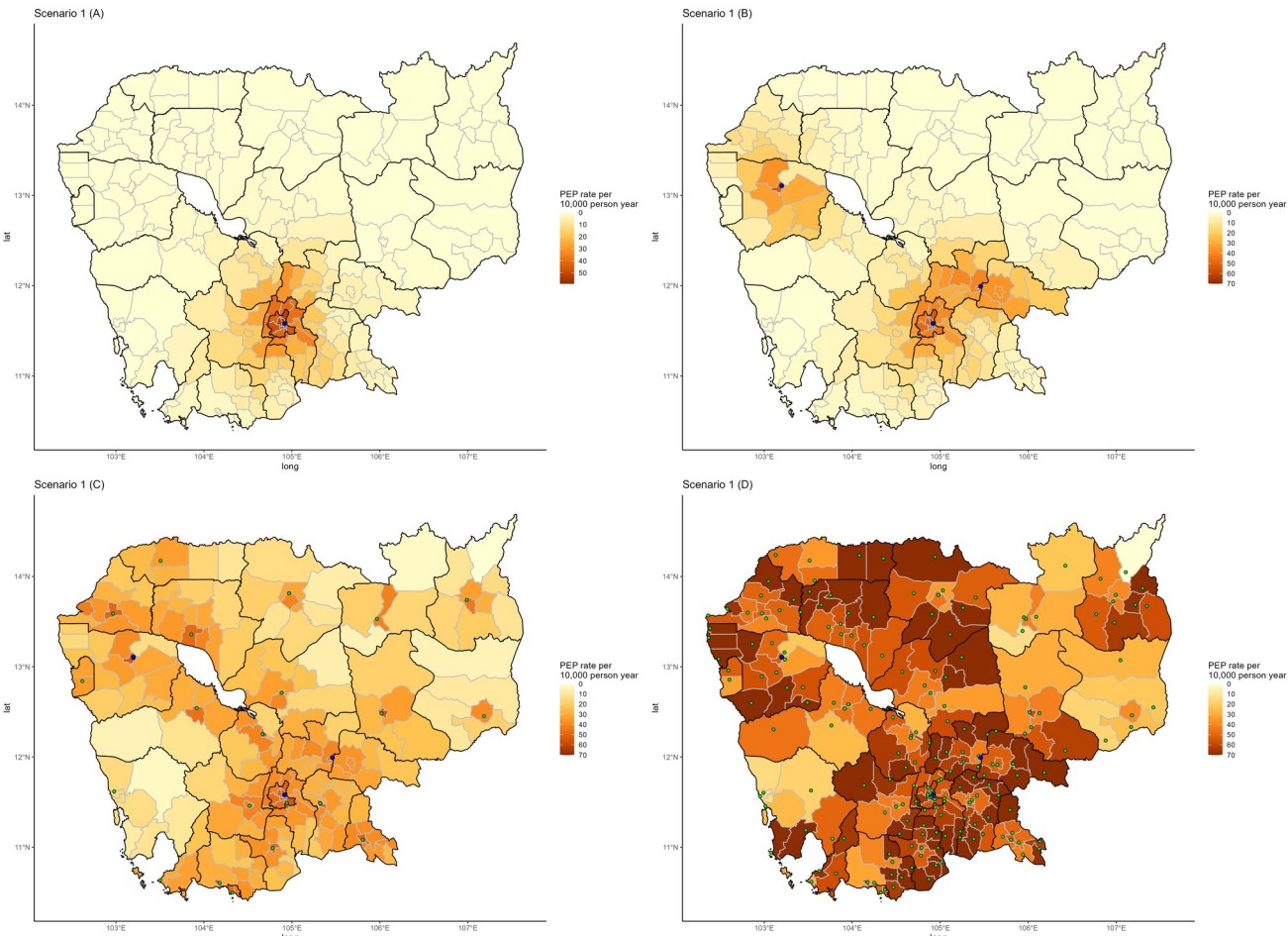

**Fig 4. Predictions of the rate of PEP patients in the population for the year 2017 based on three scenarios.** (A) Scenario 1 represents the situation prior to the opening of new centers in Battambang and Kampong Cham provinces with a single center in Phnom Penh. (B) Scenario 2 represents the current situation, with the opening of two new centers in Battambang and Kampong Cham provinces that actually opened in 2018 and 2019 respectively, bringing the total number of centers to three. (C & D) Scenario 3 and 4 represens the theoretical opening of a center in every provincial capital and district capital respectively. Blue dots represent currently existing centers as of 2020, green dots represent provincial or district capitals where future centers could be opened. Base map can be found at [45]. https://data.humdata.org/dataset/cambodia-admin-level-0-international-boundaries. Details for the corresponding license can be found at: https://data.humdata.org/faqs/licenses.

predicted patients, ranging from 193 (PEP center in Mondul Kiri) to 3,336 (PEP center in Siem Reap). Four locations led to more than 2,300 additional predicted patients: Siem Reap, Svay Rieng, Takeo and Beantay Meanchey. Another three provinces yielded increases between 1,500 and 1,900: Kampot, Kampong Speu and Kampong Thom. Results for Scenario 5 are summarized in S2 Table.

## 4. Discussion

### 4.1 PEP in Cambodia

The last estimation of the rabies burden in Cambodia was published by Ly et al. in 2009, with 800 estimated human deaths per year [6]. A recent survey performed in Battambang and Kandal Provinces showed that the yearly bite incidence remains among the highest in the world, with 2.3% in Kandal and 3.1% in Battambang respectively [10]. Thanks to the commitment of IPC to rabies burden mitigation in Cambodia, this survey provides an updated picture of the PEP needs of Cambodia. A rough extrapolation to the whole country would imply 375,000 bite injuries per year: even with an over-estimation, a large proportion of bitten people probably remain untreated. The proportion of victims who were infected by rabies remains unknown. From 2000 to 2007, PEP patient numbers were relatively stable with an average 13,020 patients per year, i.e., a rate of 10.4 patients per person-year. In 2008, this jumped to a new level of stability, with 21,100 patients a year or 15.0 patients per person-year. This major shift was associated with a shift in the geographical distribution of patients with 65% of cases originating from Phnom Penh in 2000–2007 compared to 45% in 2008–2016, suggesting improved awareness in rural provinces. Despite this, Cambodia has comparatively low PEP rates compared to neighboring countries where PEP is accessible in multiple centers. Vietnam reported 43.0 PEP patients per 10,000 person-year between 2005 and 2015 with nearly 500,000 patients in 2017. Although patients where still clustered in certain areas, access to PEP in Vietnam appeared more homogeneously distributed than in Cambodia [19,20]. However, both countries showed wide ranges of PEP rates between provinces with values ranging from 0.14 patients per person-year in Otdar Meanchey to 65.33 in Phnom Penh for Cambodia compared to a range of 2.21 to 156.27 in Vietnam. In Thailand, another country which has increased its PEP capabilities, the number of patients has increased from 90,000 in 1991 to 400,000 in 2003, which approximately represents an increase in rate from 16 to 60 patients per 10,000 person [57].

### 4.2 Accessibility and surveillance

As expected, accessibility was significantly associated with the rate at which individuals sought PEP following an animal attack. In both univariate and multivariate associations, and at both provincial and district levels, increasing travel time led to a reduction of the rate of PEP within the population. Between 2013 and 2016, 75% of PEP patients at IPC Phnom Penh came from districts with a median travel time to IPC below 60min, and 96% of patients came from districts with a median travel time to IPC below 120min. For comparison, 27% of the Cambodian population in 2016 was living in districts located within 60min of IPC and 53% in districts within 120min.

The effect of the other two variables, urban population proportion and travel time to provincial capitals, varied with the model type (univariate or multivariate). This can be explained by the correlation between the three variables. There is a very high correlation between travel time to provincial capital and travel time to IPC, with a Pearson's correlation coefficient value of 0.78 at the province level and 0.74 at the district level. Provincial capitals, including Phnom Penh where the main IPC center is located, have better access to Phnom Penh than other parts of the country as they are on major roads, thus provinces or districts with higher accessibility

to their own provincial capital tend to have higher accessibility to IPC in Phnom Penh. This explains why the univariate results show similar results between these two variables: increasing travel time to an IPC center or to the provincial capitals both decrease the PEP rates. Once we adjusted for time to IPC center, the multivariate results showed a positive association between time to provincial capital and PEP rate, suggesting that more remote rural areas have higher than expected PEP rates, which could be due to higher bite incidence in rural areas. A survey published in 2016 estimated a biting rate of 4.8 bites per 100 person-years in the rural province of Siem Reap [25] compared to 1.1 per 100 person-years in both the urban province of Phnom Penh and the peri-urban province of Kandal in a 2011 survey [24], supporting this theory. Another more recent study found similar results with bite rates of 3.1 per 100 person-years in the rural province of Battambang compared to 2.3 per 100 in Kandal [10].

Conversely, urban proportion is negatively correlated with travel time to provincial capital and, by extension, with travel time to IPC, meaning that urbanized areas are closer to provincial centers. However, this correlation is stronger at the district level (Pearson's r = -0.42) compared to the province level (Pearson's r = -0.25). Cambodia is a rural country and has only one urban province where the majority of the population lives in urban areas, Phnom Penh, which is also the smallest in area as it is limited to the city of Phnom Penh. In the 2008 census, 93.6% of Phnom Penh Province's population was living in urban areas. In comparison, the proportion of people living in urban areas varied from 1.7% to 40.4% in other provinces. Therefore, as high urban proportion is strongly correlated to being close to Phnom Penh, we would expect that they would similarly be associated with higher PEP rates. Once adjusted for travel time, the association between PEP rate and the urban population proportion remained positive but with a much smaller odds ratio, still suggesting that more urban areas have higher PEP rates. This could be explained by the fact that urban areas, especially Phnom Penh, have higher economic and development metrics, thus better access to general healthcare and information about PEP availability at IPC, as well as higher incomes [43]. Evidence of cost being a barrier to PEP can be seen in the drop in PEP rates in 2010 and 2011, following the introduction of a $10 fee for PEP at IPC, compared to 2009 when PEP was free. On the other hand, at the district level, we observed 30 highly urban districts, where more than 50% of people lived in urban areas as defined by the census. However, nearly all highly urban districts outside of the Phnom Penh area are where provincial capitals are located, thus urban proportion is highly correlated with travel time to provincial capital, and, by extension, time to IPC. In this case, adjusting for time to IPC brought the association of urban proportion and PEP rates even closer to the null. The impact of urbanization on PEP-seeking behavior should be cautiously interpreted since urbanization can serve as a proxy for socio-economic indicators and PEP awareness, which would increase expected patient numbers in urban areas. Urbanization also serves as a proxy for lower risk of rabid dog attack, which would decrease expected patients in urban areas relative to rural areas. A survey looking at awareness showed high levels of rabies awareness in both the peri-urban province of Kandal and the urban Phnom Penh, but lower awareness of IPC's existence in Kandal (12%) compared to Phnom Penh (32%) [24]. Similarly, individuals in Kandal were less likely to go to a clinic or hospital (47%) compared to Phnom Penh (66%) following a dog bite. A study performed in the province of Siem Reap showed higher education was associated with higher knowledge, and farmers had lower knowledge compared to other professions [58].

Overall, our results suggest an underreporting of dog-bite incidence, and that accessibility to a PEP center is a significant barrier for bitten individuals. This is in accordance with a previous study looking at the rate of completion of PEP regimens in Cambodia: patients living further away from IPC being less likely to complete their regimen [59].

As a tropical country, Cambodia has a dry season and wet season with a monsoon. We suspected that flooding from the Mekong River and Tonle Sap Lake during the wet season might lead to lower accessibility and thus lower rates of PEP from June through October. However, preliminary data observation showed no evidence of seasonal pattern in PEP rate nationally or at the provincial level, suggesting that the rainy season is not a significant barrier to care (S3 Fig). Tarantola et al. also found no association between the climate seasonality and the rate of PEP completion [59].

### 4.3 Expanding accessibility and predictions

When using the models to predict the impact of increased accessibility on the expected number of patients, we observed similar large increases in the number of patients at both province and district level models. In 2017, with only one center in Phnom Penh, the observed number of patients was 22,421, which is very close to our prediction of 21,712 in the district level model. In July 2018, a new center was opened in the city of Battambang and another was opened in March 2019 in the city of Kampong Cham. Our district-level model predicted some 4,010 patients at the Battambang PEP center and 4,228 at the Kampong Cham PEP center for 29,950 total patients in Cambodia. In 2019, with all three centers operational, 78,691 patients were recorded, of which 15,070 were at the Battambang center and 11,132 at the Kampong Cham center. These numbers are much higher than our predictions. However, patients at IPC in Phnom Penh also dramatically increased to 52,498. This is likely not related to any increase in accessibility but presumably to the impact of the story of a young girl bitten by a cat in December 2018 who died of rabies in February 2019. This event was widely distributed on social media and led to a dramatic increase of PEP requests in the following months [60,61]. Indeed, the number of patients at the IPC Phnom Penh reliably averaged 1,811 a month from January 2012 to November 2018, with a narrow range between 1,512 and 2,319 patients. In December 2018, patient numbers increased to 2,906 and then continued increasing to a peak of 7,593 in March before coming back down to an average of 3,397 patients per month from April 2019 through December 2020. The number of patients in 2019 represents an increase of 134% in Phnom Penh compared to 2017. If we assume a similar rate of increase nationwide due to this event, our total expected number would be closer to 33,600 patients in 2019, of which 6,400 would be in Battambang and 4,800 in Kampong Cham. Though we have few data points from the opening of the Battambang center to when these events unfolded, we do observe a similar trend. The center opened in July 2018 and only recorded 84 patients in its first two months of operation, but then averaged 172 patients per month from September through November. In December, the number of patients at the Battambang center jumped to 600 and continued increasing to a peak of 1,501 in April 2019 before coming back down to an average of 1,157 from May 2019 to December 2020. Our projection for Battambang estimated 327 patients a month, which is actually higher than what was initially observed at the Battambang center prior to the increases of December 2018, but much lower than the values that followed. As the Kampong Cham center did not open until March 2019, a similar comparison could not be conducted. As a whole, patient numbers in 2020 decreased from the surge of 2019 but remained much higher than expected with some 36,634 patients in Phnom Penh, 12,957 in Battambang, and 9,593 in Kampong Cham. However, the drop in numbers in 2020 may also be attributable to movement restrictions caused by the Coronavirus pandemic, with all three centers reaching the lowest number of patients in April 2020.

In prediction scenarios with higher accessibility to a PEP center, the difference between the province-level and district-level model predictions increased. This is likely due to the district-level model assuming a more accurate population distribution than the province level

aggregation. Within each province, the population is more likely to be concentrated within districts that are close to travel infrastructure and urban centers. The province-level model did not account for this more localized information. Scenarios representing a theoretical universal access to PEP more than doubled the expected number of patients if we assumed a center in each provincial capital, and tripled (province-level model) or quadrupled (district-level model) if we assumed a PEP center in every district. Nevertheless, even in Scenario 4, which modelled a center in each district, we still observed higher rates of PEP per population in districts near Phnom Penh compared to others. Districts of Phnom Penh and neighboring provinces tend to be smaller and have a denser road network, thus they still have higher accessibility to their own administrative centers compared to districts in the forested and mountainous regions in the southwest and northeast. Similarly, a corridor of high rates can be observed in districts surrounding the Tonle Sap lake (Fig 4C and 4D), as these are along the two main north-south highways of Cambodia and thus have better accessibility to nearby administrative centers. Even with a PEP center in each district, around 91% of the population would live within 60 minutes of a center.

Given the cost of opening new centers, it is impossible to implement in practice a center in every Cambodian district. Even a PEP center in every provincial capital is unlikely in the short to medium term. We used the district-level model with Scenario 5, with its higher spatial and demographic resolution, to identify which individual provincial capital would have the most beneficial impact. Four provinces stood out as the most effective locations to increase access to PEP: Banteay Meanchey, Siem Reap, Svay Rieng and Takeo. Opening a PEP center in one of these four provinces increased the number of yearly patients by a range of 2,300 to 3,300, and increased the population living within 60 minutes of center by a range of 503,000 to 683,000. A fifth location in Kampot would increase the number of patients by 1,800, and the population living within 60 minutes of a center by 381,000.

However, even when considering accessibility, predictions led to PEP rates that are much lower than bite rates reported from surveys. Our highest estimate projected 0.6 PEP patients per 100 person-years in 2017, against 1.1 bites per 100 person-years in Phnom Penh and Kandal provinces and 4.8 bites per 100 person-years in Siem Reap province [24,25]. These results show that even in the best-case accessibility scenario, there would still be under-reporting and under-coverage of bite injuries. This is despite the very high awareness of rabies in the population, with more than 90% of people knowing of the presence of the disease in dogs and more than 70% knowing it is fatal according to surveys conducted in Phnom Penh, Kandal, and Siem Reap Provinces [24,58]. However, the Angkor Hospital for Children in Siem Reap and the National Institute of Public Health clinic in Phnom Penh also provide PEP in Cambodia [23,26]. Thus, IPC data are not reflective of all PEP patients in Cambodia, even if these two centers combined report 3,500 patients per year, which is much lower than the average of 22,000 patients per year at IPC. Furthermore, Ponsich et al. reported that 12% of bite victims in Siem Reap received PEP from private clinics [25]. Thus, a number of bite victims seek PEP in institutions that are not captured in our data, whether they receive effective PEP or not. Next, a majority (75%) of bitten and interviewed people in Siem Reap reported using traditional treatments and a minority reported any sort of modern medical treatment (36%) [25]. Similarly, only 56% of bitten respondents in Kandal and Phnom Penh sought medical treatment and even fewer, 21%, were aware of the existence of IPC [24]. In both cases, these numbers were lower in Kandal than in Phnom Penh. The massive increase in patients in 2019 due to a media event is a clear example of the impact that media outreach can have in increasing awareness of the disease's impact and the ways of preventing it.

### 4.4 Model and data limitations

As Bayesian statistics using Markov chain Monte Carlo (MCMC) can be computationally intensive and difficult to parametrize, a method using INLA to approximate the posterior marginal distribution and developed for R was used in our study [51,53]. Studies comparing R-INLA to other regression modeling methods have shown that R-INLA can be simpler to use and quicker in computation whilst yielding similar estimates to other Bayesian or generalized linear approaches [55,62]. We believe this was the most cost-effective approach in this particular study.

However, the data used for this study had several limitations. First, the accessibility data were only available as one data point repeated over time, meaning that we could not capture the impact of the rapid improvement of Cambodian road infrastructure over the last three decades on PEP rates. We initially considered using a spatially autocorrelated random effect in the PEP model. However, because our accessibility variables were themselves spatially autocorrelated (e.g., districts close to IPC are close to other districts close to IPC and its inverse) and because of the lack of variation in accessibility over time within each geographical unit, the random effect negated the impact of accessibility as a fixed effect. Therefore, we did not use random effects in our PEP models, meaning that we could not adjust for the impact of specific geographic location beyond accessibility to IPC or its level of urbanization. This meant we could not identify areas with higher rates of PEP when adjusted for accessibility and urbanization. Furthermore, the global friction raster used to calculate accessibility relies on broad assumptions regarding mobility on a given type of terrain or infrastructure that might not be applicable to every segment of the population in every country. This might be particularly true for bite injury victims which could have impaired mobility. Moreover, travel speed depends on accessibility to a vehicle, which is limited in a country where only 5% of households owned a car and 44% owned a motorbike in 2008, though this would have likely increased since [43,63]. However, in the absence of a more locally-specific mobility study, we assumed this would still be reflective of the broad accessibility issues for the country, and that if the assumptions from the Global Malaria Atlas project introduced a bias, this bias would be mostly similar throughout the country. Nevertheless, this does create unmeasured uncertainty around our estimates.

Secondly, given the infrequency of census data, demographic data were estimated for most years using linear projections, which is a simplistic approach to demographic projections. Finally, the lack of socio-economic indicators meant that the use of urban proportion was difficult to interpret. This indicator can be used as a proxy for many factors, such as wealth, education, how dogs are maintained, or exposure risk to a rabid dog. The 2008 census included such indicators, and likely the full 2019 census will as well. However, the provisional 2019 census did not yet include this information. Similarly, we did not have detailed information about the distribution of the dog population for all provinces and districts of Cambodia, which could also be a meaningful indicator of bite risks.

Finally, the models relied on data from 2000 to 2016, making them unable to take into account the opening of centers in Battambang and Kampong Cham Provinces, which would have provided data directly measuring the impact of new center openings. Moreover, as is clearly shown by the discrepancy in numbers between projections and patient numbers actually observed in 2019, recent events have dramatically changed awareness in a way that could have lasting impact, limiting the validity of our predictions, which clearly underestimate the current situation. It can also be assumed that the COVID-19 pandemic might have major medium to long term effects on mobility and accessibility as well as on IPC and other health infrastructure, further reducing the certainty around forecasted estimates.

### 4.5 PEP in the broader context of rabies prevention

It is important to note that PEP is not the only tool available to combat rabies. However, it is the main tool currently being implemented in Cambodia on a large scale. Beyond PEP access, IPC has launched large scale education campaigns to increase awareness of the disease and how to prevent it [64]. However, it is widely recognized that the most cost-effective way to combat rabies is canine vaccination, which can be supplemented by population management strategies other than culling [16,65–69]. However there are currently no large scale canine vaccination programs in Cambodia, with control plans focused on expanding PEP accessibility, surveillance, and population awareness [70]. Nevertheless a pilot vaccination and canine demographic study has recently been conducted in two provinces of Cambodia [10]. This work is aimed at studying the feasibility and requirements of vaccination campaigns in Cambodia and will inform disease transmission and vaccination models currently under development.

## 5. Conclusion

Accessibility to a PEP center is one of the main barriers to obtaining PEP, and with only one main center in operation until 2018, large portions of the country had little access to this life-saving treatment. As a consequence, PEP rates in Cambodia varied between 9.9 and 16.2 patients per 10,000 people from 2000 to 2016. In comparison, neighboring Vietnam, which has broader access to PEP, showed rates varying from 38.9 to 65.5 per 10,000 between 2005 and 2015 [19]. Furthermore, with the centralization of most PEP at IPC and the lack of data collection from private clinics, the data collection with regard to both bite victim estimates and animal testing was highly centralized, making it difficult to establish a detailed picture of the distribution of rabies risk and burden as well as the need for PEP and canine vaccination. In 2018, IPC started expanding its capabilities in terms of PEP distribution and data collection by opening of two new centers, one in Battambang and the other in partnership with a provincial hospital in Kampong Cham. Based on our results, more centers would clearly be necessary to provide broader access to PEP, as large portions of the population remain in areas that are distant from the existing centers. This study helps identify provincial capitals that should be prioritized as most efficient locations for future centers. Two broad areas stand out as ideal future locations, one in the northwest of the country, where the neighboring provinces of Banteay Meanchey and Siem Reap both showed potential for large increases in patients, despite the fact that both are bordering Battambang Province, where a center is now located. In the south, the two neighboring provinces of Takeo and Kampot represent another area where at least one center would be advised. Finally, the province of Svay Rieng to the southeast could be seen as a possible third location.

## Supporting information

**S1 Table. Observed cumulative PEP patients by province from 2000 to 2016.**
(DOCX)

**S2 Table. Predicted accessibility to PEP centers and new PEP patients in the 21 simulations of Scenario 5 in comparison with Scenario 2.** Scenario 2 represents the current situation in Cambodia with three PEP centers available in Phnom Penh, Battambang and Kampong Cham. Scenario 5 includes 21 models that each add one center from the 21 remaining provinces to the three in existence. Scenario 5 was used to identify the best future location for adding new centers.
(DOCX)

**S3 Table. Nation-wide predictions of PEP patient numbers and rates according to four prediction scenarios.** Results from both province and district level models are presented. Scenario 1 represents the situation prior to the opening of new centers in Battambang and Kampong Cham provinces with a single center in Phnom Penh. Scenario 2 represents the current situation, with the opening of two new centers in Battambang and Kampong Cham provinces that actually opened in 2018 and 2019 respectively, bringing the total number of centers to three. Scenario 3 represents the theoretical opening of a center in every provincial capital. Scenario 4 represents the theoretical opening of a center in every district.
(DOCX)

**S4 Table. Predicted number of patients by province based on four prediction scenarios.** Results from both province and district level models are presented. Scenario 1 assumes no opening of new vaccination centers. Scenario 2 assumes the opening of two new centers in Battambang and Kampong Cham provinces that actually opened in 2018 and 2019 respectively. Scenario 3 assumes the opening of a center in every provincial capital. Scenario 4 assumes the location of a vaccination center in every district. This is a theoretical scenario used as a proxy for universal access.
(DOCX)

**S1 Fig. Urban population proportion by district for the year 2016.** Estimates based on linear projections of the urban and rural populations by district from the 1998 and 2008 census. Red dots represent the location of provincial capitals. Most urbanized districts in Cambodia are where provincial capitals are located. Base map can be found at [45]. https://data.humdata.org/dataset/cambodia-admin-level-0-international-boundaries. Details for the corresponding license can be found at: https://data.humdata.org/faqs/licenses.
(TIF)

**S2 Fig. Accessibility maps.** Travel time to point rasters for (A) IPC in Phnom Penh as was the case up to 2017, (B) all three current vaccination centers including the one in Battmabang opened in 2018 and the one in Kampong Cham opened in 2019,(C) all provincial capitals in Cambodia based on the 2010 administrative break-down and (D) all district capitals. Blue dots represent the points of interest for each maps: vaccination centers or provincial capitals. Base map can be found at [46]. https://gadm.org/download_country.html. Details for the corresponding license can be found at: https://gadm.org/license.html.
(TIF)

**S3 Fig. Monthly distribution of patients by year with genelarized additive model (GAM) smoothed curve.** GAM curve is represented with a thick blue line and uncertainty shading.
(TIF)

**S1 Data. PEP patients per province and year from 2000 to 2016 dataset.** This dataset includes a time and space aggregation of PEP patients from the initial surveillance data obtained from the IPC at the provincial level.
(CSV)

**S2 Data. PEP patients per district and year from 2013 to 2016 dataset.** This dataset includes a time and space aggregation of PEP patients from the initial surveillance data obtained from the IPC at the district level.
(CSV)

**S3 Data. Cambodia population projections per year from 1998 to 2019 dataset.** This dataset is composed of entries from the 1998, 2008 and 2019 censuses at the province and district

levels, as well as projections based on these years for the years 1999 to 2007 and 2009 to 2018. (CSV)

## Author Contributions

**Conceptualization:** Jerome N. Baron, Véronique Chevalier, Beatriz Martínez-López.

**Data curation:** Jerome N. Baron, Sowath Ly, Veasna Duong, Yik Sing Peng.

**Formal analysis:** Jerome N. Baron.

**Funding acquisition:** Jerome N. Baron, Véronique Chevalier, Sowath Ly, Beatriz Martínez-López.

**Methodology:** Jerome N. Baron, Beatriz Martínez-López.

**Project administration:** Véronique Chevalier, Sowath Ly.

**Resources:** Véronique Chevalier, Sowath Ly, Beatriz Martínez-López.

**Supervision:** Véronique Chevalier, Sowath Ly, Philippe Dussart, Didier Fontenille, Beatriz Martínez-López.

**Validation:** Jerome N. Baron, Beatriz Martínez-López.

**Visualization:** Jerome N. Baron.

**Writing – original draft:** Jerome N. Baron.

**Writing – review & editing:** Jerome N. Baron, Véronique Chevalier, Philippe Dussart, Didier Fontenille, Beatriz Martínez-López.

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
