## [Decision Letter · Decision Letter 0]

6 Aug 2021

Dear Dr Baron,

Thank you very much for submitting your manuscript "Impact of accessibility to rabies
centers on human Rabies post-exposure prophylaxis rates in Cambodia and prediction
of future needs using spatio-temporal Bayesian regression modelling" for
consideration at PLOS Neglected Tropical Diseases. As with all papers reviewed by
the journal, your manuscript was reviewed by members of the editorial board and by
several independent reviewers. In light of the reviews (below this email), we would
like to invite the resubmission of a significantly-revised version that takes into
account the reviewers' comments. 

We cannot make any decision about publication until we have seen the revised
manuscript and your response to the reviewers' comments. Your revised manuscript is
also likely to be sent to reviewers for further evaluation.

Sincerely,

Mauro Sanchez, ScD

Associate Editor

Guilherme Werneck

Deputy Editor

Reviewer's Responses to Questions

**Key Review Criteria Required for Acceptance?**

**Methods**

-Are the objectives of the study clearly articulated with a clear testable hypothesis
stated?

-Is the study design appropriate to address the stated objectives?

-Is the population clearly described and appropriate for the hypothesis being
tested?

-Is the sample size sufficient to ensure adequate power to address the hypothesis
being tested?

-Were correct statistical analysis used to support conclusions?

-Are there concerns about ethical or regulatory requirements being met?

Reviewer #1: In this manuscript, the authors investigate the relationship between
travel time to rabies treatment centers and the rate that individuals receive
post-exposure prophylaxis (PEP) to prevent rabies in Cambodia. They first use a
Bayesian regression model to fit the relationship between travel time and PEP rate
using data reported from 2000-2016. They find an inverse relationship between travel
time (accessibility) and PEP rate, as expected. Then they use this fitted
relationship to estimate how much PEP would increase if additional testing
facilities had been built in each province or district in 2017 (under 3 different
scenarios). They conclude by identifying the 5 provincial capitals where they would
recommend adding an additional treatment center.

What I liked about the manuscript was the creative approach the authors use to
identify high-risk areas in the absence of reliable human disease risk or burden
data. There are also several aspects of the rationale and methods that need to be
addressed or clarified before publication, which I have described below.

1) Throughout the manuscript it was hard to follow what the authors meant by the
different scenarios, and which ones they were referring to (table ST2 helped). I
suggest (1) including at least one main table or diagram that describes the
analytical approach clearly and (2) being explicit throughout the results and
discussion section about which scenario and model (district or province) you are
referring to. It would also be helpful to explicitly describe which model/scenario
is used to recommend new treatment centers (and why that one was chosen).

2) It would be helpful to have more details on how new treatment centers were placed
in scenarios 2-4. In scenario 3 and 4, were the centers from scenario 2 that
currently exist included (it appears this way from the figures)? In scenario 4, were
the centers randomly place or were they placed within the largest city or existing
health facilities/clinics? Do lines 215-216 refer to scenario 3, and was this the
scenario used for final conclusions? Please clarify these points in the text.

3) Why were Bayesian regression models chosen for identifying locations for a new
treatment center? Since the goal is reducing travel time, why not just put a new
center in a location that would maximally reduce travel time in either total hours
or population-weighted in person-hours? Would this change the results? It looks like
this might be what was done in “Section 3.2 Population accessibility”, but that
section includes no corresponding methods or figures so it’s hard to follow.

4) What was the purpose of estimating the number of animals tested? This analysis
doesn’t directly address the objectives of the study and isn’t included in the
discussion. I suggest either removing it or making the rationale for doing it very
clear (and then also including it in the discussion and conclusions).

5) On lines 517-520 you mention that there was no evidence for seasonality. Can you
show this data in an SI figure or include a reference? You could also consider
explicitly fit a non-linear model to confirm that seasonality isn’t captured here,
and/or consider including a metric of seasonality in a model to test this (for
example, including a monthly rainfall fixed effect). Though, I do not expect this to
impact the overall conclusions.

Reviewer #2: Methods

The methods chosen seem to have been carefully considered and I support the decision
to use an INLA model, because of the widely available options to incorporate spatial
and temporal dependency of the data. The authors managed to explain the four
different scenarios very well and have scientifically sounds approach. However, I
feel that the description is not always clear and the objectives become a little bit
fuzzy after the very clear introduction. If indeed the goal of this paper is to
estimate the spatial distribution of the risk of dogbite among all provinces and
districts of Cambodia, then the explanation of the methods would have to be
restructured. 

I would consider merging the estimates of dogbite patients and the estimates of
positive animal samples so that you can say something about the proportion of rabid
dogbites among people bringing in a sample along with their need for PEP. 

In addition, the methods lack an explanation on data cleaning or incomplete bite
reports. I would expect that not all patients were bitten by dogs or animals capable
of transmitting rabies, so this raises the question how many reports had to be
removed because of that. In addition, I was wondering how many bite reports couldn’t
be linked to the right administrative boundaries because of incompleteness or
misspelling. Maybe this could be elaborated in the Supplementary material. 

Population data is the main denominator for the outcomes of your estimates, make sure
to use one that is reliable across all years. I would also suggest comparing the
estimates made in this study to the widely available open datasets from for instance
WorldPop or the CIESN/facebook team. This would give you more precise input data
about the location of your data for the eventual demographic estimates. I would also
suggest you to have a look at the R package wopr (https://github.com/wpgp/wopr) to support the estimates you make for
the population data as described in lines 162-165. Find below some additional points
of feedback:

Line 216-218: “The accessibility rasters were aggregated for each province and
district polygon by extracting the median travel time from patient residence
province or district to PEP center or provincial capital value.” 

This sentence is a bit hard to follow, I would suggest rewriting to: The
accessibility measure was aggregated by means of extracting the median travel time
per province or district. This could ultimately be linked to the patient’s record
(i.e. the district or province of residence). 

Paragraph 2.2.1: I would advise you to use a constrained (using building footprints)
gridded population dataset as input data (https://www.worldpop.org/methods/top_down_constrained_vs_unconstrained).
It allows you to have more accurate information on the actual location of population
and thus the population living beyond the travel time catchments you are indicating
(i.e 60 and 120 min). The calculations you make involve two aggregated measures
(namely aggregated travel time and aggregated population counts) while in fact you
have finer scale spatial data available to deal with the calculations in a more
precise manner. 

Paragraph 2.2.3: This paragraph is not very clear to me and my feedback links back to
the more general point raised before “to consider merging the estimates of dogbite
patients and the estimates of positive animal samples”. To me it does not seem
logical to predict the number of dogs that would be brought in for testing. This
would make sense if your goal is to scale up your FAT testing capacity, but not to
predict the risk of rabid dogbites. In my opinion it would make more sense if you
use the number of positively tested samples among the patients that brought in a
sample, as a proxy for the number of rabid dog bites and thus potentially averted
deaths by the use of PEP. In a recent article written by Rajeev et al (2021),
dogbites in Madagascar were used in a decision tree model to estimate the risk of a
rabid dogbite in a Bayesian MCMC model. If this is also your aim, I would consider
redoing a part of your model here and re-describe your goal and method here.

Line 252-254: Sentence “In an aggregated logistic regression..” is a bit challenging
to read I would suggest to rewrite.

**Results**

-Does the analysis presented match the analysis plan?

-Are the results clearly and completely presented?

-Are the figures (Tables, Images) of sufficient quality for clarity?

Reviewer #1: (No Response)

Reviewer #2: Results

The results presented reflect the methods introduced before and are clearly lined out
in text and figures. However, I find paragraph 3.3. challenging and would suggest to
restructure and rewrite this. Below you can find a more detailed description.

Line 279-280: Were all these patient reports complete? How many were available at the
province level and how many at the district level? 

Paragraph 3.2: For the statistics about access to PEP I would highly suggest you to
use a gridded population dataset instead of the aggregated counts. See example
Rajeev et al. (2021). This would really increase the validity of your estimates.
Proportionally calculating people’s access to care without taking into account their
exact location leads to too much uncertainty of your estimates. 

Line 355-356: Can you compare the provincial and district models? Since not all input
data was collected at the district level, I would also expect a less powerful model
output on district level because not all bite reports can be brought back to this
level. 

Paragraph 3.3: It’s a well written paragraph but it remains unclear to me what
exactly you want to model and to communicate to your audience. I think the main goal
is to understand the distribution of rabies risk and thus the need to improve the
spatial availability of PEP in high risk regions. To me it seems therefore that you
would like to report 1) on the number of patients estimated to have missed out on
PEP because of poor access, which would be your predicted number of patients minus
the reported number of animal bite patients 2) the estimated risk of a patient being
bitten by a rabid animal, which would be the probability of an animal to be rabid
times the number of reported animal bites. 

The increase in PEP patients to me seems a bit as an indirect measure. Because it
represents the estimated total number of patients based on access to PEP centers
over the last years. So reporting onthe regions where the difference between
reported and predicted bite victims is highest would be the most important outcome,
that is where you are missing access.

**Conclusions**

-Are the conclusions supported by the data presented?

-Are the limitations of analysis clearly described?

-Do the authors discuss how these data can be helpful to advance our understanding of
the topic under study?

-Is public health relevance addressed?

Reviewer #1: 6) In either the introduction/rationale or discussion/future work it
would be helpful to understand whether building new treatment centers is the best
way to prevent deaths due to rabies, compared with using those funds to make
treatments free, increase awareness, vaccinate dogs, or some combination thereof,
and how you might integrate that information with the results from this type of
analysis.

Reviewer #2: Discussion & conclusion

Again, well written and provides a good overview of the results but would be more
tangible with reports on measures as “averted deaths”, “human burden of diseases”,
“proportion of rabid positive bites”

Line 431: “bite injuries” instead of “bites injuries”

Paragraph 4.2: The authors mention an important limitation of the accessibility layer
used in paragraph 4.4, reflecting the static usage of the layer which is likely to
have changed over the years because of improvements in the road network. However,
the way accessibility is measured in this study brings another uncertainty to the
estimates. The widely used malaria atlas friction layer makes large scale
assumptions on road networks and travel speeds on different landcover types, which
in fact might be different for your target population, namely people who seek access
to care. In the paper published by Weiss et al (2018) https://www.nature.com/articles/nature25181. You can find back the
travel speeds applied to each country. I would highly suggest you to have a look and
check whether they seem reliable for your target population.

**Editorial and Data Presentation Modifications?**

Reviewer #1: - In the title, “prediction of future needs” is misleading since that is
not actually what is done in this study, and I suggest removing or rephrasing.

- In Fig 2, it would be helpful to see results in both absolute and relative terms
since the numbers, especially since the overall are so small.

- If Fig 3b, the “missing” label seems incorrect.

- In Fig 4, please include a panel for scenario 4.

- A lot of my comments require adding additional clarity or details. Since the
manuscript is already long, some of the more descriptive text and discussion could
be made more concise or removed.

Reviewer #2: Figures:

Overall great figures, but would suggest to clip the estimated travel times to the
country borders.

**Summary and General Comments**

Reviewer #1: (No Response)

Reviewer #2: Thank you for the opportunity to review this interesting paper on the
effect of accessibility to PEP providing health centers on bite patient estimates to
support rabies control strategies. The authors have written a very good piece and
the scope of the work fits well with the PLoS NTD audience. Rabies research faces a
lot of challenges regarding data availability and the authors applied thorough
methods to estimate the number of actual bite victims in comparison to the reported
bite victims in Cambodia. However, some of the methods and results have to be
adapted to further improve the validity of the outcomes.

My main concerns are related to the population data in combination with the
accessibility measures. Aggregated population counts in combination with aggregated
measures of accessibility, create more uncertainty in the predictions and outcomes
mentioned in the results. Since the travel time rasters are gridded I would highly
suggest the authors to also use a gridded population dataset that provides
information on the location of the population, like the WorldPop constrained
dataset. 

In addition, the way the results are written make it hard to follow the actual goal
of the paper, which in my opinion is to estimate the risk of a rabid animal bite. I
would consider a stronger emphasis on the number of patients estimated to be missed
by the PEP clinics (estimated patients – reported patients) and the number of
estimated rabid positive bites (number of positive samples * estimated animal
bites). A visual representing the aimed outcomes or a decision tree model would
maybe help guide the reader in understanding the results better. I provide more
detail below. 

The paper would also greatly benefit from a detailed read through to improve grammar.
There are quite some inconsistencies with missing words or words that are plural and
miss and s at the end. 

Finally, the accessibility measures that are adapted from the global friction
surfaces as provided by the Malaria project introduce uncertainty on actual travel
times of the target population, because speeds on roads and landcover types have
been set-up in a global studuy, not always reflecting the local context
realistically. 

Yet, the paper provides useful insights into an important NTD in a context that lacks
recent research. The paper has high potential for publication when the feedback
provided is taken into consideration. 

All the best with the suggestions and I’m looking forward to reading an improved
version of the paper!

PLOS authors have the option to publish the peer review history of their article
(what does this mean?). If published, this will
include your full peer review and any attached files.

If you choose “no”, your identity will remain anonymous but your review may still be
made public.

**Do you want your identity to be public for this peer review?** For
information about this choice, including consent withdrawal, please see our
Privacy Policy.

Reviewer #1: No

Reviewer #2: No
---

## [Decision Letter · Decision Letter 1]

27 Jan 2022

Dear Dr Baron;

Thank you very much for submitting your manuscript "Impact of accessibility to rabies
centers on human rabies post-exposure prophylaxis rates in Cambodia and
identification of optimal future center locations using spatio-temporal Bayesian
regression modelling" for consideration at PLOS Neglected Tropical Diseases. As with
all papers reviewed by the journal, your manuscript was reviewed by members of the
editorial board and by several independent reviewers. The reviewers appreciated the
attention to an important topic. Based on the reviews, we are likely to accept this
manuscript for publication, providing that you modify the manuscript according to
the review recommendations. 

After reviewing the revised version you have submitted, we would like to ask you to
address the points raised by the reviewers to allow your work to be deemed suitable
for publication. I suggested that your go through all comments provided below and
provide special consideration to changing the title your manuscript as indicated by
reviewer 1 and carry out a thorough revision of grammar and spelling.

We look forward to receiving your revised version and congratulate the team on a
careful work in addressing previous comments.

Sincerely,

Mauro Sanchez, ScD

Associate Editor

Guilherme Werneck

Deputy Editor

Dear Dr Baron and co-authors;

After reviewing the revised version you have submitted, we would like to ask you to
address the points raised by the reviewers to allow your work to be deemed suitable
for publication. I suggested that your go through all comments provided below and
provide special consideration to changing the title your manuscript as indicated by
reviewer 1 and carry out a thorough revision of grammar and spelling.

We look forward to receiving your revised version and congratulate the team on a
careful work in addressing previous comments.

Reviewer's Responses to Questions

**Key Review Criteria Required for Acceptance?**

**Methods**

-Are the objectives of the study clearly articulated with a clear testable hypothesis
stated?

-Is the study design appropriate to address the stated objectives?

-Is the population clearly described and appropriate for the hypothesis being
tested?

-Is the sample size sufficient to ensure adequate power to address the hypothesis
being tested?

-Were correct statistical analysis used to support conclusions?

-Are there concerns about ethical or regulatory requirements being met?

Reviewer #1: I thank the authors for adding Table 1 in response to my comments. It is
very helpful, but I suggest editing the text for "scenario 5" to reflect the
description in the text of the methods because it's currently unclear. For example,
the description could read something like: "21 scenarios with four PEP centers,
where each scenario includes the three current PEP centers (Phnom Penh, Battambang
and Kampong Cham) and one additional center added in each of the 21 provincial
capitals." In addition, I'm unsure what "as scenario 1,2, and 3" means for the
Source.

Lines 314-315. "The result of this analysis was not significant". Please be more
specific about what you were specifically testing here, and what was "not
significant".

Reviewer #2: Thank you very much for the opportunity to review a revised version of
this manuscript. The paper was already very interesting and good at the first round
of review, but the authors did an outstanding job at further improving the study. I
would like to thank the authors for their detailed feedback on the comments raised
and have no additional technical comments for consideration in this new round of
revisions. The authors improved the explanation of the different scenarios
substantially by adding an extra table and also further highlighted the limitations
of the input data. I find the paper fit for publication after some small
improvements of grammar and spelling. I think the paper would even further improve
after a native english speaker could read over it.

**Results**

-Does the analysis presented match the analysis plan?

-Are the results clearly and completely presented?

-Are the figures (Tables, Images) of sufficient quality for clarity?

Reviewer #1: I'm a bit confused about why Scenario 5 gets introduced first in section
3.2. Shouldn't lines 460 to 471 belong in the last paragraph of the results where
Table ST4 is referenced? If I am missing something here then it would be helpful to
have further description of the rationale of this scenario in both sections of the
results.

Reviewer #2: Yes

**Conclusions**

-Are the conclusions supported by the data presented?

-Are the limitations of analysis clearly described?

-Do the authors discuss how these data can be helpful to advance our understanding of
the topic under study?

-Is public health relevance addressed?

Reviewer #1: Their conclusions are supported and in their revisions the authors have
added additional limitations and public health context important for understanding
the implications.

Reviewer #2: Yes

**Editorial and Data Presentation Modifications?**

Reviewer #1: The title of the manuscript is still a bit misleading. In this paper,
the authors examine associations between accessibility and yearly PEP patient rates
(and use that fitted relationship to predict optimal new PEP center locations),
rather than causally assessing the "impact". I suggest replacing "impact" with
"association between". Alternatively, the title could be rephrased to read something
along the lines of: "Accessibility to rabies centers and human Rabies post-exposure
prophylaxis rates in Cambodia: a spatio-temporal analysis to identify optimal
locations for future centers".

Reviewer #2: I think the paper would even further improve by having a native speaker
go over the text once. I started to notice some tiny mistakes in the abstract (see
below), but then focussed more on the bigger picture of the paper and the content. 

Title

“in Cambodia and identification of optimal future center locationsusing
spatio-temporal Bayesian regression modelling” add space between locations &
using. 

Abstract

Line 34: “to inform the selection future PEP center locations in high-risk areas”
suggest to change sentence to “to inform the selection of future PEP center
locations in high-risk areas”

Line 35: “accessibility to rabies centers on the yearly rate of PEP patient in the
population and generate a risk map” suggest to add and “s” behind patient > PEP
patients 

Line 38: “patients as outcomes” remove “s” behind outcome. 

Line 39: “Secondary exposure variables consisted in” change to “consisted of”

**Summary and General Comments**

Reviewer #1: Overall, the authors have done a thorough job of responding to my
comments and my remaining comments are mostly editorial.

Reviewer #2: Strong study with a very important societal impact in a rabies endemic
country. Authors have been able to produce reliable estimates of human rabies and
dogbite burden that can aid prioritization and allocation of resources. Figures and
tables support the findings of the studies and the improvements made by the authors
help guide the reader through the paper. Well done and I look forward to seeing the
rest of the research coming alive!

PLOS authors have the option to publish the peer review history of their article
(what does this mean?). If published, this will
include your full peer review and any attached files.

If you choose “no”, your identity will remain anonymous but your review may still be
made public.

**Do you want your identity to be public for this peer review?** For
information about this choice, including consent withdrawal, please see our
Privacy Policy.

Reviewer #1: No

Reviewer #2: No

Figure Files:

Data Requirements:

Reproducibility:

References

---

## [Editor Report · Decision Letter 2]

12 May 2022

Dear Dr Baron,

We are pleased to inform you that your manuscript 'Accessibility to rabies centers
and human rabies post-exposure prophylaxis rates in Cambodia: a Bayesian
spatio-temporal analysis to identify optimal locations for future centers' has been
provisionally accepted for publication in PLOS Neglected Tropical Diseases.

Best regards,

Guilherme L Werneck

Deputy Editor

I want to thank the authors for providing adequate answers to the reviewers and
revising the manuscript accordingly.

---

## [Editor Report · Acceptance letter]

8 Jun 2022

Dear Dr Baron,

We are delighted to inform you that your manuscript, "Accessibility to rabies centers
and human rabies post-exposure prophylaxis rates in Cambodia: a Bayesian
spatio-temporal analysis to identify optimal locations for future centers," has been
formally accepted for publication in PLOS Neglected Tropical Diseases.

Best regards,

Shaden Kamhawi

co-Editor-in-Chief

Paul Brindley

co-Editor-in-Chief
